# STIM Proteins: An Ever-Expanding Family

**DOI:** 10.3390/ijms22010378

**Published:** 2020-12-31

**Authors:** Herwig Grabmayr, Christoph Romanin, Marc Fahrner

**Affiliations:** Institute of Biophysics, Johannes Kepler University Linz, Gruberstrasse 40, 4020 Linz, Austria; Herwig.Grabmayr@jku.at

**Keywords:** STIM1, STIM2, isoforms, Orai, CRAC, SOCE, CC1, NMR, structure, simulation

## Abstract

Stromal interaction molecules (STIM) are a distinct class of ubiquitously expressed single-pass transmembrane proteins in the endoplasmic reticulum (ER) membrane. Together with Orai ion channels in the plasma membrane (PM), they form the molecular basis of the calcium release-activated calcium (CRAC) channel. An intracellular signaling pathway known as store-operated calcium entry (SOCE) is critically dependent on the CRAC channel. The SOCE pathway is activated by the ligand-induced depletion of the ER calcium store. STIM proteins, acting as calcium sensors, subsequently sense this depletion and activate Orai ion channels via direct physical interaction to allow the influx of calcium ions for store refilling and downstream signaling processes. This review article is dedicated to the latest advances in the field of STIM proteins. New results of ongoing investigations based on the recently published functional data as well as structural data from nuclear magnetic resonance (NMR) spectroscopy and molecular dynamics (MD) simulations are reported and complemented with a discussion of the latest developments in the research of STIM protein isoforms and their differential functions in regulating SOCE.

## 1. Introduction

Chemical elements in their ionic form are indispensable factors for the correct function of vital cells and organisms. Among the various ions that have been selected in the course of evolution to be involved in living organisms, calcium (Ca^2+^) occupies a special place. Besides the enormous amounts found in bones and teeth, calcium plays an outstanding role as a second messenger in every cell [1,2,3,4]. In a resting cell, calcium is present in a very low cytosolic concentration. For intracellular calcium-dependent signal transduction, an increase of the cytosolic calcium concentration is necessary. This is achieved by calcium influx from the extracellular space or by depletion of the intracellular calcium store in the endoplasmic reticulum (ER) [4]. Among the different calcium-selective transmembrane proteins, the pathway of store-operated calcium entry (SOCE) plays an important role. Two key proteins, stromal interaction molecule (STIM) and Orai, form the calcium release-activated calcium (CRAC) channel system that mediates SOCE and is responsible for regulated calcium influx in many cell types [5,6,7,8]. Orai is the calcium-selective channel in the plasma membrane (PM) and STIM is the calcium sensor in the ER membrane. Ligand binding to the extracellular surface of the cell leads to cytosolic activation of phospholipase C (PLC), which in turn cleaves the head group of a specific phospholipid to form cytosolic inositol trisphosphate (IP3) and membrane-bound diacylglycerol (DAG). IP3 diffuses throughout the cytosol to IP3 receptors in the ER membrane and elicits depletion of the store. As a result, the ER luminal calcium concentration decreases dramatically [2,4,9] This is the step that represents the STIM-activating signal [10,11,12,13]. STIM has, among other elements, an EF hand in its ER luminal N-terminus with which the calcium concentration in the ER can be sensed [14,15]. Lowering the ER calcium concentration results in dissociation of calcium from the STIM EF hand and thus in a conformational change of the STIM N-terminus [10]. The signal propagates across the transmembrane (TM) domain to the cytosolic C-terminus of STIM [16]. A cascade of conformational changes occurs at the STIM C-terminus, resulting in oligomerization and spatial extension of the protein [16,17,18,19,20,21,22]. The extended STIM protein translocates to the cell periphery and interacts directly with the PM-resident protein Orai [8,23,24]. The physical coupling to Orai occurs mainly at its C-terminus; however, interactions with Orai loop2 and N-terminus are also involved in the correct gating of the channel [25,26]. In its monomeric form, Orai has 4 TM domains, a cytosolic N- and C-terminus, and a cytosolic loop2 between TM2 and TM3. Six Orai monomers form the functional hexameric Orai channel [27]. Six TM1 domains constitute the calcium-selective channel pore, which is separated from the hydrophobic milieu of the PM and from the TM4 domains by a ring consisting of TM2 and TM3. The Orai N-terminus merges into the TM1 and the C-terminus merges into the TM4 [8,25,27]. The importance of SOCE for physiological cytosolic calcium homeostasis and the calcium-dependent function of critical cell-biological processes is underlined by several gain- (GoF) and loss-of-function (LoF) mutations within STIM and Orai. GoF mutations raise the intracellular calcium concentration by eliciting constitutive CRAC channel activation as well as SOCE. This leads to a clinical continuum including disease phenotypes termed York platelet syndrome, Stormorken syndrome, and tubular aggregate myopathy. Pathological manifestations thereby include (but are not limited to) myopathy, thrombocytopenia, miosis, ichthyosis, and dyslexia. In contrast, LoF mutations abolish CRAC channel activation. The absence of SOCE results in severe combined immunodeficiency, autoimmunity, ectodermal dysplasia, and muscular hypotonia [28,29].

In this review, we focus on the growing family of STIM isoforms, which are specifically expressed in various cell types. Key regulatory domains of the protein involved in stabilizing the STIM resting state and domains involved in protein activation are described. The focus is on competitive interactions of cytosolic domains of STIM. The recently published functional data as well as structural data from nuclear magnetic resonance (NMR) spectroscopy and molecular dynamics (MD) simulations have been essential in advancing and complementing the characterization of STIM [15,21,22,30]. Moreover, we report results of ongoing functional investigations that are based on these novel NMR data.

## 2. STIM Proteins

STIM is a dimeric type I single-pass TM protein which is mainly anchored in the ER membrane [5,31,32,33] and to some extent in acidic stores [34] as well as the PM [35,36,37]. It generally consists of an N-terminal ER luminal portion and a larger C-terminal portion in the cytosol that are connected by a TM domain (Figure 1) [8]. STIM possesses two main functions: on the one hand, it is a precise sensor of the calcium concentration within the ER lumen; on the other hand, it couples to and gates calcium-selective Orai channels in the plasma membrane [6,7]. In order to perform these two tasks, STIM is equipped with several specialized domains spread across its N- and C-terminal portions [7,38]. There are two homologous STIM proteins called STIM1 and STIM2, each having different isoforms that have been characterized since the discovery of the protein family. For STIM1, these include STIM1 Long (STIM1L) and the recently discovered STIM1A (Figure 1a) [39,40]. Two studies by Miederer et al. and Rana et al. in 2015 revealed a total of three STIM2 isoforms: STIM2.1 (or STIM2β), STIM2.2 (or STIM2α), and STIM2.3 (Figure 1b) [41,42]. In this nomenclature, the conventional isoform of STIM2 is termed STIM2.2 and will be used hereafter. All isoforms will be discussed in detail after a comprehensive introduction to STIM proteins that includes the latest developments in the field and a brief report of ongoing functional investigations.

The STIM N-terminal ER luminal portion harbors a canonical EF hand (cEF) as well as a noncanonical EF hand (nEF) motif [5,32]. A sterile alpha motif (SAM) domain is found further downstream before the TM domain marks the end of the N-terminal portion (Figure 1) [46,47]. The cytosolic C-terminal portion is considerably larger than the N-terminal ER luminal portion. It includes several coiled coil (CC) domains. The first one, termed CC1, immediately connects to the TM domain. It is followed by a further two CC domains known as CC2 and CC3. Due to their involvement in the activation of Orai channels, CC2 and CC3 are often together referred to as CRAC-activating domain (CAD) [48] or STIM-Orai-activating region (SOAR) [49]. Downstream of CAD/SOAR, further domains for inactivation (ID-STIM) and microtubule (MT) end binding (EB) are found. A polybasic lysine-rich domain (PBD) sits at the outermost C-terminus, allowing it to interact with negatively charged phospholipids in the PM (Figure 1) [8,48,49,50,51]. Additionally, the CC1 domain is further subdivided into three α-helices named α1, α2, and α3 [20,21,22,52,53,54], while CAD/SOAR is separated into four helical regions named Sα1, Sα2, Sα3, and Sα4 [52,55].

## 3. STIM N-Terminus

The STIM calcium sensor within the ER lumen precisely and sensitively reacts to changes of the ER Ca^2+^ level in a concentration range between 100 and 400 µM [23,56]. The Ca^2+^ ion concentration for half-maximal activation has been determined to amount to approx. 200 µM for STIM1 and 400 µM for STIM2.2 (cf. Section 5 for further details on the differences between the STIM homologs) [23,57]. The cEF thereby is the effective calcium-binding domain while the nEF provides stabilization [46,47]. Upon a decrease in the ER luminal calcium concentration, calcium dissociates from the cEF. This loss of calcium together with a nearby variable random coil sequence destabilizes the whole EF–SAM complex, which elicits subsequent STIM oligomerization [46,47,58]. It remains unclear how many Ca^2+^ ions are bound by the STIM ER luminal domains. Pioneering studies have found that the STIM cEF consists of two helix–loop–helix motifs, which is in agreement with the consensus structure of EF hands. Furthermore, it was reported that EF–SAM contains a single Ca^2+^ ion coordinated by the loop of the first helix–loop–helix in cEF [10,46,47]. However, recent studies have indicated that a total of 5–6 Ca^2+^ ions may be bound to STIM at the same time. Since STIM is a dimeric protein, this would mean that 10–12 Ca^2+^ ions are bound per dimer in the inactive state [10,59,60,61]. The locations of these putative binding sites remain to be elucidated, though their calcium binding seems to be energetically coupled to the EF hand [59].

The N-terminal ER luminal domains of both STIM1 and STIM2.2 have been resolved in high resolution by NMR spectroscopy [46,47]. The structures provide insight into the inactive state of STIM which is associated with a high ER luminal Ca^2+^ ion concentration. Detailed structures of the activated STIM1 and STIM2.2 N-termini are still lacking, although some insight has been provided by a recent study. Enomoto and coworkers performed X-ray crystallography on the ER luminal STIM domains of *C. elegans* and found high structural conservation to the human paralog in a high-calcium environment [62]. More importantly, solution NMR spectroscopy in a low-calcium environment unveiled an α-helix-to-β-strand transition of the SAM C-terminal helix as well as conformational exchange of the EF hand between folded and unfolded states. These results provide insight into the structural rearrangements required to elicit STIM oligomerization and conformational change [62]. The availability of structures for the N-terminal ER luminal STIM domains has also led to the inclusion of MD simulations as a complementary approach to standard cell-based assays in STIM research. Specifically, recently published MD simulations of the STIM1 N-terminus have enabled the prediction of hitherto unknown binding sites for Ca^2+^ ions [13]. Even though these MD simulations were carried out at supraphysiological Ca^2+^ concentrations and are pending future experimental confirmation, results like these could be the key to finally map the predicted amount of 5–6 Ca^2+^ ions to the inactive STIM N-terminus [10,13,59,60,61]. MD simulations have also been successfully applied to study ER luminal STIM1 GoF mutations associated with tubular aggregate myopathy [63,64], which have also been the subject of investigations at the cellular and whole organism levels [13,65].

When the ER calcium stores are full, the EF–SAM domains of the STIM dimer are spaced apart and thereby prevent spontaneous activation [66,67]. This is supported by an early study on STIM1, which reported that artificial crosslinking of the ER luminal domains leads to oligomerization as well as Orai1 channel activation irrespective of the ER calcium store level [23]. Recently, the mechanism responsible for transmitting the activating signal from the ER lumen to the cytosol was unveiled. In their study, Hirve et al. revealed an underlying zipping coiled coil mechanism [12]. Interaction of the EF–SAM domains elicits structural rearrangements of the TM domains according to this model. This leads to coiled coil formation and propagation of the signal to the cytosolic C-terminal portion [12,16].

## 4. STIM C-Terminus

In store-replete conditions, the STIM cytosolic C-terminal portion is in a tightly packed, folded state. This state is stabilized by an interaction between CC1 and CAD/SOAR, which is also known as coiled coil “clamp” due to the nature of the interacting domains [16,18,19,20]. Several point mutations have been discovered that hint towards CC1α1 and CC3 being key for the formation of the clamp [20,21]. While there is no definitive proof for the number and identity of the interacting side chains, L248, L251, L258, and L261 within CC1α1 are well-known residues that disrupt the clamp upon mutation. As possible interaction partners, residues within CC3 were suggested, which can also lead to clamp release upon mutation. However, whether these actually are the directly interacting amino acids has not yet been clarified. The hypothesized CC3 counterparts are L416, V419, and L423 [16,18,19,20,68]. Judging from the evidence, these leucines and valines could form the clamp through intramolecular hydrophobic interactions, which are sufficiently strong to stabilize the STIM1 inactive state while still allowing a conformational switch to occur. This hypothesis is supported by the fact that mutation of just one of these side chains, either in CC1 or CC3, to a hydrophilic serine precludes clamp formation and leaves STIM1 in a permanently extended state, as well as by hydrophobic mutations such as R426L, which strengthen the clamp interaction to the point where no STIM1 activation may occur anymore [18,20,21]. In addition to the essential STIM1 CC1α1–CC3 clamp formation in the resting state, both CC1 as well as CC3 are responsible for STIM1 oligomerization processes, which become active after ER store depletion. It has been explicitly shown that homomerization of the CC1α1 domain occurs in the course of STIM1 activation. This molecular event is mechanistically coupled to the release of the CC1α1–CC3 clamp. Subsequently, additional regions within CC3 oligomerize; this promotes the formation of larger STIM1 clusters [17,20].

In the context of the CC1α1–CC3 clamp, the residue R304, whose mutation to tryptophane (R304W) causes the Stormorken syndrome [69,70,71], was unexpectedly found to play a major role as well. Located at the very end of the CC1α2 helix and thus some distance away from the CC1α1 and CC3 residues that are thought to form the coiled coil clamp, STIM1 R304W is constitutively active and locks STIM1 in the active state [21]. Thorough mechanistic investigations have revealed that the mutation increases homomerization of CC1α1 within the STIM1 dimer, which normally is a hallmark of activated STIM1 [12,21]. In addition to this, the tryptophane residue at position 304 has been shown to increase the rigidity of the flexible loop between CC1α2 and CC1α3 by extending the CC1α2 helix. This restriction of the degrees of freedom together with increased homomerization subsequently inhibits CC1α1–CC3 clamp formation [21,22].

Very recently, a novel structural resolution of the STIM1 CC1 domain monomer by NMR spectroscopy was reported by Rathner et al., revealing a three-helix bundle of the α1, α2, and α3 helices, which are arranged in an antiparallel fashion and connected by two flexible loops (Figure 2a, left panel) [22]. This result is in sharp contrast to the CC1 X-ray crystallography structure that was previously reported by Cui and coworkers in 2013 [72]. The 2013 structure exhibits a fully stretched conformation, which is quite unusual. However, a CC1 M244L + L321M double mutant has been utilized to create a crystal, which is a likely cause for the discrepancy between the two structures [22,72]. Indeed, Rathner et al. report that in their NMR structure, both M244 and L321 are located in the positions important for interhelical interactions within CC1. Furthermore, they observed partial pre-activation of Orai1 currents in live-cell patch clamp electrophysiology with STIM1 M244L + L321M and thus suggested that the double mutation used by Cui and coworkers has a GoF impact on the functional behavior of STIM1 in line with the stretched CC1 crystallographic structure. The STIM1 CC1 NMR structure has also revealed two novel intramolecular interhelical nuclear Overhauser effect (NOE) contacts between the CC1α1 and CC1α2 helices. One contact is formed between L258 and L261 of α1 to I290 and A293 of α2, while another one was found between L248 and L251 of α1 and L300 and L303 of α2 (Figure 2a, right panel) [22]. Interestingly and as already mentioned above, L248, L251, L258, and L261 all disrupt the CC1α1–CC3 clamp upon mutation [16,18,19,20,68]. Since this match clearly hints towards an effect of the CC1α1–CC1α2 interactions on the coiled coil clamp, Rathner et al. disrupted the interactions by targeted double point mutations of the hydrophobic CC1α2 residues to hydrophilic serines. This strategy ensured preservation of the principal CC1α1–CC3 interaction. Functional characterization using patch clamp electrophysiology and various Förster resonance energy transfer (FRET) assays on both full-length STIM1 and different STIM1 protein fragments revealed a distinctive regulatory impact. Most prominently, the STIM1 I290S + A293S double mutant exhibited significantly slowed current activation kinetics while STIM1 L300S + L303S led to a significantly reduced Orai1 current activation plateau. Even more impressively, the constitutively active Stormorken STIM1 R304W mutant was returned to a STIM1 wild type-like behavior when combined with either of the double point mutations [22]. Using the FRET-derived interactions in a restricted environment (FIRE) method [20], the authors were able to determine a significantly increased interaction between the CC1 and CC3 domains as the underlying mechanism. CC1α1 and CC3 are the key domains keeping STIM1 in the resting state by formation of their clamp interaction. Thus, it seems likely that weakening or disturbing the interactions between L258/L261 and I290/A293 as well as between L248/L251 and L300/L303 (Figure 2a, right panel) by mutation of the CC1α2 residues confers enhanced freedom to the CC1α1 residues for establishing the CC1α1–CC3 clamp interaction. Hence, the resulting coiled coil clamp is now stronger, stabilizing the STIM1 resting state and overruling the activating effect of the Stormorken R304W mutation. Overall, the reported STIM1 CC1 domain monomer NMR structure has considerably advanced the understanding of the STIM1 activation cascade by revealing that a fine-tuned balance between CC1 interhelical interactions and contacts to the CC3 domain of CAD/SOAR is involved in the control of CRAC channel activation [22].

The prominent effect of the newly identified CC1 interhelical contact mutations on the Stormorken STIM1 R304W mutant observed with patch clamp electrophysiology (Figure 3a,d) and various FRET assays [22] raises the question whether these mutations can also overrule other STIM1 GoF mutants that are neither directly part of the CC1α1–CC3 binding interface nor of the interhelical contacts and thus principally do not preclude clamp formation. Here, we report the first experimental results for STIM1 C227W, a TM domain mutant [16], as well as for STIM1 D247C, which is situated in the STIM1 coiled coil zipping region [12] as part of ongoing functional studies. In patch clamp electrophysiology experiments (according to the method published in [22]) with HEK293 cells co-expressing CFP–Orai1 and YFP–STIM1, the C227W mutant showed robust constitutive Orai1 current activation. The combination of C227W with the I290S + A293S double mutation to create STIM1 C227W + I290S + A293S did not elicit any significant change in the constitutive Orai1 current activation behavior (Figure 3b). Similarly, STIM1 D247C induced strong constitutive Orai1 currents that were not altered in experiments using the triple mutant STIM1 D247C + I290S + A293S (Figure 3c). Thus, I290S + A293S was only able to significantly abolish or reduce constitutive currents induced by STIM1 R304W and not by the other two tested GoF STIM1 mutants C227W and D247C (Figure 3d).

The patch clamp electrophysiology results shown in Figure 3 point towards distinct mechanistic effects being responsible for the behavior of the tested STIM1 GoF mutants. Due to their positions in the TM domain and in the CC1α1 domain, both C227W as well as D247C possibly are able to interfere earlier in the STIM1 activation cascade compared to R304W and/or more potently enforce the active conformation. Thus, the strengthened CC1α1–CC3 clamp resulting from I290S + A293S might be energetically too weak to compete or not able to be formed any longer. Like R304W, both C227W and D247C are capable of pushing STIM1 into the activated extended state even in store replete conditions. A zipper-like mechanism is initiated hereby, which is associated with the well-known CC1α1 homomerization. Hypothetically, in the course of this interaction, the CC1α1 domains get closer to each other, resulting in the interface for CC3 not being perfectly available and consequently the CC1α1–CC3 clamp no longer being able to endure. There is strong evidence from Fahrner et al. supporting this view, demonstrating that, unlike R304W, mutation of L251S in CC1α1 disrupts both the CC1α1–CC3 interaction as well as the CC1α1 homomerization in FIRE experiments, which again hints towards distinct molecular GoF mechanisms [21]. Consequently, L251 has at least a triple function: it is involved in the maintenance of the CC1α1–CC3 clamp in quiescent STIM1, in the CC1α1 homomerization of activated STIM1, and in the interaction with L300 within the CC1α1–CC1α2 interhelical contacts (Figure 2a, right panel) [22]. This strongly supports the hypothesis that L251 within CC1α1 has multiple possible interaction partners depending on the orientation of CC1α1.

A comparison of the stabilizing effects of the two CC1 interhelical contact mutations reveals considerable differences. Interestingly, STIM1 L300S + L303S significantly reduces the Orai1 current activation plateau compared to wild type STIM1 both in the presence and absence of R304W, while STIM1 I290S + A293S has no effect on maximum currents and still exhibits enhanced kinetics of activation in the presence of R304W [22]. It might therefore be worthwhile to also combine the C227W and D247C GoF mutations with the L300S + L303S double mutation and probe for possible effects on Orai1 current activation. Additionally, a STIM1 quadruple mutant combining both interhelical contact double serine mutations to yield STIM1 I290S + A293S + L300S + L303S represents an exciting experimental target since additive effects on the CC1α1–CC3 clamp strength are conceivable. Further studies are required to determine whether L300S + L303S or the quadruple mutation I290S + A293S + L300S + L303S are able to compensate the activating effects of C227W and D247C. Moreover, the investigation into the effects of the CC1 interhelical contact mutations will need to be expanded to further STIM1 GoF mutants. Taken together, mechanistic understanding of different STIM1 GoF mutants is set to provide detailed insights that could prove pivotal for the elucidation of the STIM1 activation cascade.

In terms of structural resolutions, another STIM1 NMR fragment that partially overlaps with the CC1 structure has been reported by Stathopulos and coworkers in 2013 [54]. Termed CC1_[TM-distal]_–CC2, the fragment comprises the CC1α3 and CC2 domains (Figure 1a) and was resolved both in the presence and absence of Orai1 C-terminus. The authors were able to observe differences in double α-helix intertwining between the Orai1-free and Orai1-bound states. Furthermore, they suggest a mechanism for STIM1–Orai1 C-terminus coupling in the CRAC complex assembly. The reported STIM1–Orai1 interactions were experimentally confirmed by patch clamp electrophysiology [54]. Further along the C-terminus, the STIM1 SOAR domain has also been resolved by X-ray crystallography in 2012 (Figure 1a and Figure 2b) [52]. A triple mutant (L374M, V419A, and C437T) was used to create the crystal in this case. SOAR was reported to have an R-shape created by the antiparallel orientation of CC2 and CC3 and divided into four helices: Sα1, Sα2, Sα3, and Sα4. The first and the last of these helices correspond to CC2 and CC3 with Sα2 and Sα3 being two small helices in between them. Furthermore, the authors identified SOAR as a dimer and found interaction between the N- and C-terminal regions of the monomers (Figure 2b) [52].

While Sα1 and Sα4, i.e., CC2 and CC3, have been the subject of detailed investigations for quite some time [17,54,74,75,76], only a few studies involving the Sα2 helix of CAD/SOAR have been published to date [55,77,78][55,77,78]. Wang et al. compared the CAD/SOAR domain sequences of STIM1 and STIM2.2 and, while they are highly conserved, determined that their functional distinction stems from the Sα2 helices which differ by just a single amino acid. This substitution of a phenylalanine residue (F394) in STIM1 to a leucine residue (L485, including the 87 amino acid signal peptide insertion) in STIM2.2 causes a severe decrease in Orai1 channel gating efficacy. Additionally, their experiments revealed that mutation of the phenylalanine residue to a dimensionally similar but polar histidine (F394H) in full-length STIM1 precludes both binding to and gating of Orai1 channels. Thus, being positioned at the apex of the CAD/SOAR domain, F394 was found to represent a potential STIM–Orai coupling position (Figure 2b) [55]. In a follow-up study, the authors showed that CAD/SOAR heterodimers of the wild type and F394H are able to normally bind to and gate Orai1 channels, indicating unimolecular coupling between STIM1 and Orai1 [77]. Accordingly, a second follow-up study showed that CAD/SOAR dimers can mediate crosslinking between Orai1 channels, while heterodimers of the wild type and F394H fail to do so [78]. These results underline the importance of the F394 residue within the CAD/SOAR domain of STIM1. Moreover, in 2020, new evidence has suggested binding of F394 to the Orai C-terminal M4 extension helix, narrowing down the putative interaction counterpart of F394 on the Orai channel side [79].

When compared to the Sα2 helix, even less has been known about the Sα3 helix until recently. A novel study focusing on this segment (Figure 2b) has been published in 2019. Butorac, Muik, and coworkers found that Sα3 is essential for the activation of Orai1 as it contributes to the transmission of the STIM1 signal to Orai1 [80]. While STIM1 Sα3 point or deletion mutants are still able to bind to Orai1, they fail to trigger activation of CRAC currents which separates STIM1 coupling to Orai1 from gating of Orai1. Furthermore, the authors applied cysteine crosslinking and biochemically mapped the interaction of Sα3 with Orai1 to the TM3 domain of the latter. The interacting residues were thereby found to be L402 within STIM1 and E166 within Orai1. In patch clamp electrophysiology, strong CRAC currents were activated upon crosslinking of STIM1 L402C and Orai1 E166C, underlining the functional significance of the discovered interaction that has been termed STIM1–Orai1 gating interface (SOGI) [80]. The characterizations of both the F394 residue and of the SOGI mark important steps towards clarifying the molecular choreography of the CRAC channel complex. Nevertheless, additional structural elucidation of the CRAC channel complex as a whole is required to definitively confirm the interactions that have been proposed from the results of functional investigations.

Towards the end of the STIM C-terminus, only comparatively small modulatory domains are found (Figure 1). The first after CAD/SOAR is ID-STIM, the so called inactivation domain. A negatively charged region, ID-STIM was first reported in 2009. As its name suggests, ID-STIM is required for calcium-dependent inactivation (CDI) of the CRAC channel [81,82,83]. The exact mechanism of action of ID-STIM remains unknown and CDI can occur without it to some extent, though it was reported that three negatively charged aspartic acid residues (D476, D478, and D479) are critical for generating the full amount of CRAC channel CDI. Additionally, ID-STIM was shown to be strongly associated with Orai1 W76, as mutation of this residue completely disabled the ID-STIM function as well. ID-STIM and Orai1 W76 thus seem to act in concert to help CRAC channels progress from a “residual” inactive state to the full extent of inactivation [84]. Going further along the STIM sequence, the EB domain follows. This domain allows STIM to attach to the tips of growing MTs by interaction with EB proteins, a class of MT tip-binding proteins [51,85]. STIM1 uses interaction with EB1 to move rapidly through the cell, although it does not seem to be required for SOCE since the latter is not inhibited by knockdown of EB1. Rather, recent evidence suggests that binding to EB1 actually seems to attenuate the ability of STIM1 to migrate to ER–PM junctions and interact with Orai1 [86,87]. Accordingly, it has been shown that STIM release from MTs is elicited by Ca^2+^ store depletion [88,89]. The PBD is the final C-terminal STIM domain. As mentioned before, it permits STIM to interact with negatively charged phospholipids in the PM [8,48,49,50,51]. The domain does not seem to be critical for SOCE, although it supports pre-clustering of STIM1 and thus increases the efficacy of STIM1-mediated Orai1 activation [66,90]. It has to be noted that these results were obtained in overexpression studies and that PBD-mediated pre-clustering might have a more pivotal function at endogenous protein levels where STIM1 dimers might otherwise be too sparsely populated in order to efficiently find and oligomerize with each other after Ca^2+^ store depletion [38].

## 5. Differences between STIM1 and STIM2.2

STIM2.2 is highly similar to its homolog STIM1 (Figure 1), but still harbors some differences that distinguish it from its more prominent counterpart. Most importantly, STIM2.2 possesses a reduced affinity for calcium, making it more sensitive towards changes in the ER luminal Ca^2+^ level [47,56,91]. This stems from different degrees of structural EF–SAM domain stability of the STIM homologs which become apparent upon comparison [47,56,57]. As a result, it pre-localizes to ER–PM junctions near the resting level of the calcium store and fine-tunes the rate of Ca^2+^ entry in order to regulate and maintain the intracellular calcium homeostasis that is critical for cellular survival [56]. To achieve this, STIM2.2 triggers small but sustained SOCE that compensates for slight store depletion and thereby precludes activation of STIM1 [58]. Only if there is a strong decrease in ER luminal Ca^2+^, STIM1 gets activated and triggers considerably larger, but transient SOCE to swiftly return the calcium store back to the resting level [50,56,58]. When aligning the sequences of STIM1 and STIM2.2, deviations start to appear downstream of CAD/SOAR [92], although small differences within the domain seem to be crucial for their distinct function. Besides L485 decreasing Orai1 channel gating efficacy (cf. Section 4), another residue within CAD/SOAR of STIM2.2 (E470, again including the 87 amino acid signal peptide insertion) is responsible for weaker coupling to Orai1 according to a recent study [91]. These differences likely function to prevent STIM2.2-mediated Ca^2+^ overload. In total, STIM2.2 is 61 amino acids longer than STIM1 (Figure 1) [93]. It has also been reported that STIM2.2 forms heterodimers with STIM1, which increases the latter’s recruitment to ER–PM junctions and facilitates its activation [94,95]. Overall, STIM2.2 seems to fulfill a complementary role in the control of intracellular calcium homeostasis. This is supported by the fact that, in sharp contrast to STIM1, there are no known human diseases originating from STIM2.2 GoF or LoF mutations [96]. Nevertheless, new evidence has shown that STIM2.2 performs diverse and versatile functions in both the nervous system as well as the immune system and is slowly outgrowing its “housekeeping gene” reputation. Due to evidence for a close cooperative relationship between the STIM homologs, their physiological role might prove difficult to understand if they continue to be viewed as separate entities [96]. This becomes especially true when one begins to also consider the growing list of known STIM1 and STIM2.2 splice variants, which will be introduced in the following sections. Together, heterodimerization and alternative splicing strongly amplify the functional toolkit of STIM proteins for the shaping and modulation of SOCE.

## 6. STIM1 Isoforms

In 2011, the first STIM variant produced by alternative splicing was discovered to be a novel longer variant of STIM1 [39]. Due to its increased length compared to conventional STIM1, it was named STIM1L, which is an abbreviation for STIM1 Long. STIM1L results from the alternative splicing of exon 11, which leads to the incorporation of an additional 106 residues in the STIM1 C-terminal region that act as an actin-binding domain (ABD, Figure 1a). The latter allows STIM1L to permanently form clusters and interact with Orai1 channels [39]. Unlike STIM1, the expression of STIM1L is limited to brain and heart of mice, neonatal rat cardiomyocytes, as well as human skeletal muscle cells [39,97,98]. Due to its behavior and expression, STIM1L has been linked to rapid activation of SOCE in skeletal muscle cells, which occurs within less than one second. This is in stark contrast to other cell types, where the activation of SOCE may require several minutes [99]. In addition to rapid SOCE activation, STIM1L was further identified as a trigger for repetitive cytosolic Ca^2+^ signals [39]. For its function, STIM1L was initially suggested to rely on permanent interaction with Orai1 even at resting conditions with full Ca^2+^ stores. The association of STIM1L with actin filaments was proposed to be the stabilizer for this interaction, as actin depolymerization is sufficient to perturb STIM1L–Orai1 clusters at rest [39]. A follow-up study by Sauc et al. found that STIM1 is able to induce expansion of the cortical ER in response to Ca^2+^ store depletion via a mechanism that requires binding of the lysine-rich C-terminus to phosphoinositides while STIM1L lacks this ability [100]. This results in recruitment of Orai1 to large ER–PM junction clusters in the presence of STIM1 while these clusters are much smaller in the presence of STIM1L. The study also found that STIM1L did neither colocalize with Orai1 with full Ca^2+^ stores nor mediate rapid activation of SOCE, although STIM1/STIM1L double knockout (DKO) murine embryonic fibroblast (MEF) cells were used instead of skeletal muscle cells [100]. Thus, cell type-specific effects may be responsible for these contrasting findings. Most likely, additional proteins are pivotal for STIM1L function in skeletal muscle cells, as co-expression of STIM1 and STIM1L in DKO MEF cells also could not recapitulate rapid SOCE activation in these cells [100]. Future studies may elucidate the involvement of one or more members of the expanding class of auxiliary CRAC proteins in the function of STIM1L [101,102,103,104].

Interestingly, STIM1L has also been reported to associate with transient receptor potential canonical 1 (TRPC1) and TRPC4 channels [105,106]. Knockdown of TRPC1 and TRPC4 in human skeletal muscle cells, i.e., myotubes, was shown to reduce SOCE by half as well as significantly delay its onset with knockdown of STIM1L resulting in similar effects. Missing STIM1L negatively affected myoblast differentiation and led to the formation of smaller myotubes. Conversely, overexpression of STIM1L resulted in the development of larger myotubes. TRPC1, TRPC4, and STIM1L hence appear to be indispensable for SOCE and normal differentiation in human myotubes [105]. Another study published in 2019 corroborates the STIM1L–TRPC1 interaction and shows that STIM1L preferentially activates TRPC1 while being less efficient at gating Orai1 [106] Additionally, an interaction of STIM1L with TRPC3 and TRPC6 has been shown [97]. While having no effect on the expression level of the channels, STIM1L binds more abundantly to TRPC3, TRPC6, and Orai1 compared to STIM1 in co-immunoprecipitation experiments [97]. Clearly, additional investigation of STIM1L in connection with Orai1 and TRPC channels is required to gain a better understanding of the interplay between these proteins.

In 2020, Knapp and coworkers reported several more STIM1 splice variants containing a novel exon and/or being N-terminally truncated [40]. Their study focuses on a variant that they named STIM1A, which is generated by an additional exon A between exons 10 and 11 (12 if exon A is also numbered) of STIM1 and is not truncated. Inclusion of this newly discovered exon translates to the insertion of a novel 31-residue-long domain A within the STIM1 C-terminus in the immediate vicinity of the negatively charged ID-STIM domain (Figure 1a). STIM1A was found to be expressed in many tissues, but the highest levels were detected in testes, heart, kidney, and astrocytes. Domain A seems to be decisive for localization to unique adhesion junctions and to specialized membrane retrieval sites that are known as tubulobulbar complexes in testes [40]. On the functional level, STIM1A reduces SOCE and CRAC currents in astrocytes in a dominant-negative fashion despite there being no discernible difference in clustering and interaction with Orai1. A conserved phenylalanine–serine–aspartic acid (FSD) motif was found to be decisive for this reduction. Single alanine point mutations within the FSD motif (S502A, D503A) were sufficient for STIM1A to revert back to STIM1 behavior. The authors therefore speculate that domain A prevents a stabilizing interaction between the extended transmembrane Orai1 N-terminal (ETON) region of Orai1 and the STIM1 CAD/SOAR region, which is required for full Orai1 gating. In this model, the alanine point mutants may elicit a structural change that restores accessibility of the interacting upstream CAD/SOAR residues [40]. Knapp and coworkers conclude that different cell types use alternative splicing as (I) a targeting switch to target STIM1 variants to specific intracellular contact sites and (II) to regulate SOCE according to the requirements of those contact sites [40].

## 7. STIM2.2 Isoforms

The first of two additional STIM2.2 variants known to date, STIM2.1, was found to harbor an insert consisting of 8 residues (VAASYLIQ) in the CC2 domain of CAD/SOAR (Figure 1b). An additional exon 9 encodes the insert, which disrupts the binding to Orai1 [41,42]. Miederer et al. report that the expression of STIM2.1 is ubiquitous, though its abundance compared to STIM2.2 is dependent on the type of cells. In naïve T cells, high levels of STIM2.1 were detected with the expression levels of STIM2.1 and STIM2.2 being similar. While STIM2.2 is a promoter of SOCE, STIM2.1 was shown to fulfill an inhibitory role. Accordingly, the overexpression of STIM2.1 diminished SOCE in CD4^+^ T cells, while its knockdown had the opposite effect [41]. A number of hypotheses concerning the exact mechanism that is responsible for turning STIM2.1 into a SOCE inhibitor have been developed in the past few years. Since it was shown that there is only weak interaction between STIM2.1 and Orai1 in FRET or puncta formation assays and SOCE inhibition without a stronger interaction seems unlikely, Rana et al. have indicated heteromerization between STIM2.1/STIM1 or STIM2.1/STIM2.2 as an underlying mechanism. This is supported by evidence of heterodimerization between STIM1 and STIM2.2 [94,95]. These interactions would ensure recruitment of STIM2.1 to Orai1 and overcome its intrinsically low affinity for the channel complex [42]. Once the heterodimers are recruited, different mechanisms for SOCE inhibition are conceivable: on the one hand, passive inhibition could result from the occlusion of STIM-binding sites on Orai1 or the sequestration of STIM1/STIM2.2 in heterodimers by STIM2.1. In these cases, the number of actively bound CAD domains would be reduced and the activation of Orai1 channels would be limited as a consequence. On the other hand, STIM2.1 could actively transmit an inhibitory signal through interaction with Orai1 or Orai1–STIM1/STIM2.2 complexes [42]. Another possibility has been shown by Zhou and coworkers. Interestingly, they observed that STIM1–STIM2.1 CAD/SOAR heterodimers retain the ability to fully activate Orai1, but are defective in mediating channel crosslinking. According to these results, STIM2.1 inhibition of SOCE would solely be elicited by the absence of Orai1 channel crosslinking, which is thought to increase the efficacy of Orai1 activation [78].

Recent results from structural characterization studies on the Orai-activating small fragment (OASF, Figure 1) level have revealed a decisive impact of the STIM2.1 insert [107]. While not inducing a global conformational change, VAASYLIQ in the CC2 domain reduces the overall α-helicity of OASF, which exerts a destabilizing effect. The exposed hydrophobicity as well as conformational sensitivity of OASF to high concentrations of Ca^2+^ are additionally increased by the insert. Remarkably, these effects on the OASF domain were not limited to the STIM2.1/STIM2.2 OASF sequence context but also occurred when integrating the insert into the STIM1 OASF sequence. In line with these results, the STIM2.1 insert also precluded SOCE in full-length STIM1. The CC2 VAASYLIQ insertion therefore likely causes a dysfunctional OASF domain independent of the STIM2.2-specific context. As a result, the coupling to the Orai1 C-terminal cytosolic helices is disrupted, which prevents Orai1 channel activation [107]. Taken together, heterodimerization of STIM2.1 with STIM1 or STIM2.2 is likely responsible for the inhibitory effect of STIM2.1, as it has been shown that both CC2 helices are required to form the pocket that binds two Orai1 C-terminal helices [54]. Structural changes in one CC2 helix could thus collapse the whole Orai1 C-terminal helix-binding pocket [107]. However, it remains to be verified whether STIM2.1 inhibition is passive or active and whether there is more than one effect.

STIM2.1 has also been suggested to play a role in human heart failure. Its expression significantly decreases in end-stage heart failure, which is indicative of SOCE stimulation. This stimulation may then contribute to the development of cardiac hypertrophy [108]. Additionally, involvement of STIM2.1 in myogenesis through the control of cell cycle arrest has been reported. The latter is achieved by STIM2.1-mediated inhibition of SOCE blocking the promotion of cell proliferation, which transitions cell fate from proliferation to differentiation [109].

Not much is known about the second STIM2 splice variant STIM2.3 (Figure 1b). It is generated by the inclusion of an alternative exon 13. This alternative exon leads to an upstream end of translation and a transcript that is 444 base pairs (= 148 amino acids) shorter. Expression of STIM2.3 seems to be limited and its messenger RNA could not be detected in lymphocytes, where both STIM2.1 and STIM2.2 are expressed. At present, the function of STIM2.3 remains unknown [41].

## 8. Perspective

Although an increasing number of cell type-specific splice variants of STIM1 and STIM2 are being discovered and analyzed, STIM1 continues to be the focus of research due to its dominant role in strong Orai activation. Structural insights gained by NMR spectroscopy and X-ray crystallography as well as MD simulations allow for improved understanding of the STIM1 domains and their interactions in the course of store-dependent conformational changes of the protein. Additional analyses of existing GoF and LoF STIM1 mutants in the nature help to describe the physiologically relevant wild type STIM1 protein. Following the discovery of the minimal STIM1 fragment (CAD/SOAR) that is capable of activating Orai, it has been the slightly larger OASF fragment consisting of CC1 and CAD/SOAR that allowed more mechanistic regulatory insights. After exploring the intramolecular CC1α1–CC3 interaction in the STIM1 resting state and the subsequent intermolecular CC1α1–CC3 homomerization in the STIM1 active state, respectively, further regulatory domains could be identified, especially within CC1. However, a clear picture of STIM1 activation is still missing. Snapshots of the STIM1 activation process can be shown hypothetically, but they represent only fragments of the complete picture. Interdisciplinary research combining biophysics, structural biology, and theoretical computer-based methods offers progress. Ultimately, it is a matter of presenting the STIM1 activation cascade in the right light in order to understand how STIM1 can enter various energetically similar conformations in order to move in the right spatiotemporal order from one extreme, the tight state, to the other extreme, the fully activated state.

For 10 years, it has been well-known that in a resting cell, the STIM1 protein is present in a closed conformation with the cytosolic clamp interaction between the CC1 and CC3 domains constituting a dominant factor in the realization of this conformation. However, it is still unclear which amino acids face each other in CC1 and CC3. Here, future research requires structural elucidation of the closed STIM1 conformation. However, protein purification and aggregation is a considerable challenge, which is one reason as to why this has not yet been successful. Still, with a combination of MD simulations and functional experiments, strong evidence for the real STIM1 closed conformation can be obtained. The same applies to the representation of how STIM1 and Orai interact with each other. For more than 10 years, it has been known that STIM1 couples to the Orai C-terminus. There are structural approaches to describe the interaction, but the fragments used may be too small to correctly represent the entire STIM–Orai coupling process. Structural biology is challenged to elucidate larger interacting STIM–Orai protein fragments. Over the years, interactions from STIM to the Orai N-terminus or Orai loop2 have been identified in addition to the Orai C-terminus. Even direct interactions between the Orai N-terminus and loop2 cannot be excluded in the context of interaction with STIM. From the STIM1 point of view, there will be further investigations, especially of Sα2 in addition to Sα3, to understand how they are involved in STIM1 conformational switch and interaction with Orai cytosolic domains. Initially, MD simulations coupled with functional experiments will help in this context, in anticipation of upcoming NMR spectroscopy and X-ray crystallography structures of STIM and Orai.

## Figures and Tables

**Figure 1 ijms-22-00378-f001:**
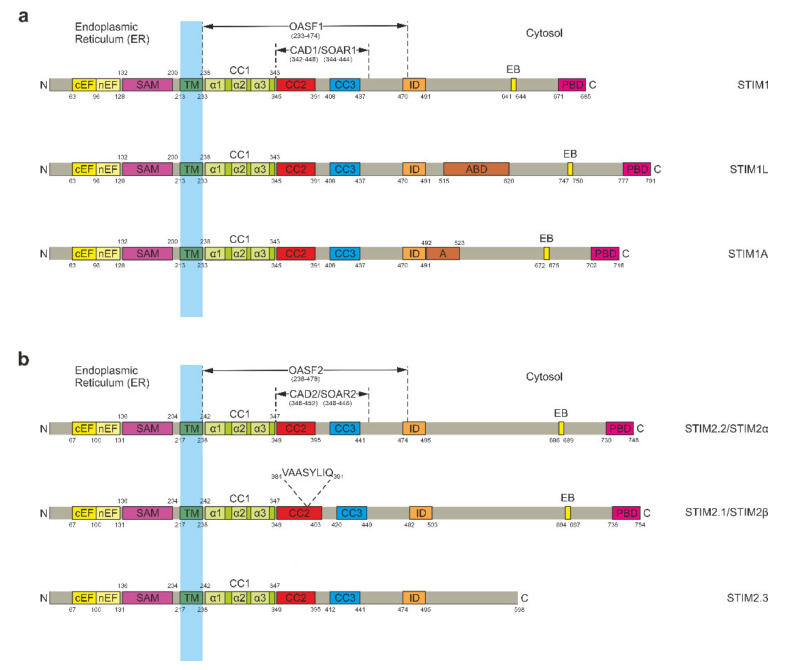
Domain structure of stromal interaction molecule (STIM) proteins. (**a**) Primary structure of STIM1 and its isoforms STIM1 Long (STIM1L) and STIM1A. Functionally relevant domains within the endoplasmic reticulum (ER) luminal portion include the canonical (cEF) and noncanonical (nEF) EF hands as well as the sterile alpha motif (SAM). Downstream of the transmembrane domain (TM), the cytosolic portion contains three coiled coil (CC) domains commonly known as CC1, CC2, and CC3 with CC1 being further subdivided into α1, α2, and α3. The C-terminal fragment spanning all three CC domains is termed Orai-activating small fragment (OASF). Another fragment comprising CC2 and CC3 is named CRAC-activating domain (CAD) or STIM-Orai-activating region (SOAR). Further C-terminal domains include the inactivation domain (ID or ID-STIM), the microtubule end-binding domain (EB), and the polybasic domain (PBD) at the outermost C-terminus. STIM1L and STIM1A feature the same general structure but each harbor an additional C-terminal domain inserted downstream of the ID domain by alternative splicing. STIM1L thereby includes an actin-binding domain (ABD) while STIM1A possesses an insert designated as domain A. (**b**) Primary structure of STIM2.2/STIM2α and its isoforms STIM2.1/STIM2β and STIM2.3. For reasons of simplicity, the 87 amino acid N-terminal signal peptide insertion of STIM2.2 and its isoforms was omitted from this display [43,44,45]. Due to the high degree of similarity between STIM1 and STIM2.2, their functional domains are essentially equivalent. Alternative splicing leads to inclusion of a small 8 amino acid insert (VAASYLIQ) within the CC2 domain of STIM2.1 and to an upstream end of translation in case of STIM2.3, shortening the protein by 148 amino acids. Otherwise, both STIM2.1 and STIM2.3 correspond to STIM2.2.

**Figure 2 ijms-22-00378-f002:**
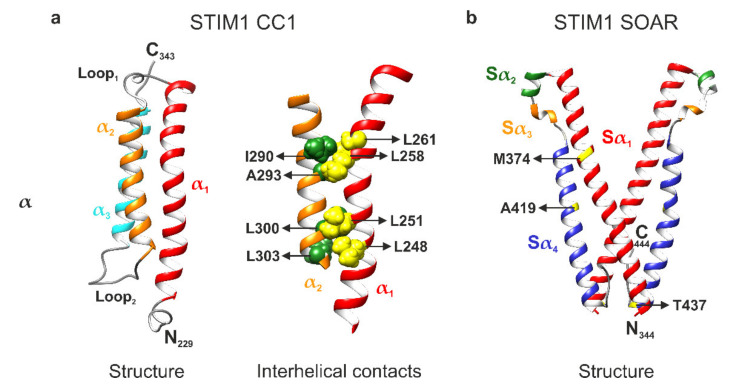
Structural resolutions of the stromal interaction molecule 1 (STIM1) C-terminus. (**a**) Left: nuclear magnetic resonance (NMR) structure of the STIM1 coiled coil 1 (CC1) domain monomer (PDB: 6YEL). The α1, α2, and α3 helices form a three-helix bundle arranged in an antiparallel fashion that is connected by two flexible loops. Right: interhelical contacts between the α1 and α2 helices. The interacting residues of α1 (L248, L251, L258, and L261; colored in yellow) and α2 (I290, A293, L300, and L303; colored in green) are shown in space-filling representation. (**b**) X-ray crystallography structure of the STIM-Orai-activating region (SOAR) dimer (PDB: 3TEQ). The domain is comprised of four helices named Sα1 (corresponding to CC2), Sα2, Sα3, and Sα4 (corresponding to CC3) and shows an R-shape created by the antiparallel orientation of Sα1 and Sα4. The crystal structure contains several point mutations (L374M, V419A, and C437T) whose positions are indicated. Molecular graphics were created with UCSF Chimera (v1.14) [73].

**Figure 3 ijms-22-00378-f003:**
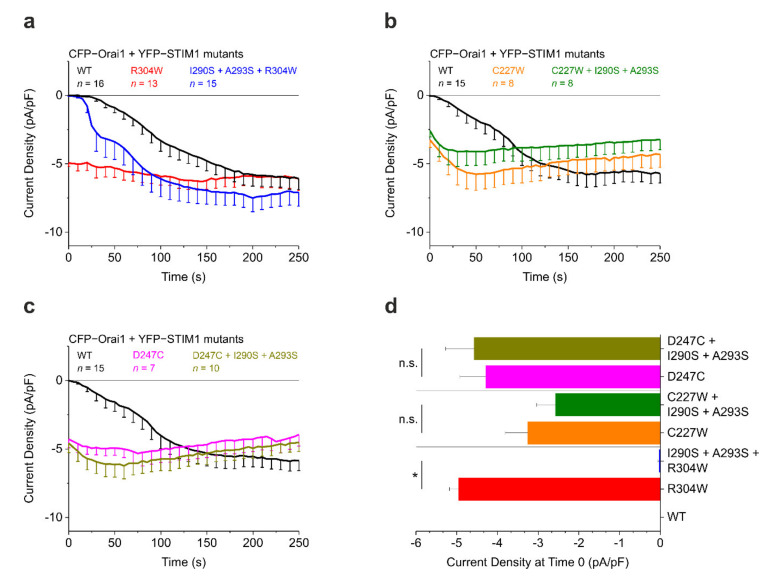
Effect of coiled coil 1 (CC1) interhelical contacts on stromal interaction molecule 1 (STIM1) gain-of-function (GOF) mutations. (**a**) Patch clamp recordings showing current activation (current density, pA/pF) of N-terminally tagged CFP–Orai1 co-expressed with YFP–STIM1 R304W ± I290S + A293S. Adapted from [22]. (**b**) Same as in (**a**), but for YFP–STIM1 C227W ± I290S + A293S. (**c**) Same as in (**a**), but for YFP–STIM1 D247C ± I290S + A293S. (**d**) Comparison of current activation levels at Time 0. HEK293 cells were used for all experiments. Student’s two-tailed t-test was employed for statistical analyses with differences considered statistically significant at *p* < 0.05. Asterisks (*) indicate significant difference (n.s., not significant). Data represent mean values ± SEM.

## Data Availability

All data are contained within the article.

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
