# Peer review of "STIM Proteins: An Ever-Expanding Family"

_ijms, 2020, doi:10.3390/ijms22010378_

Round 1

Reviewer 1 Report

Herwig et al., provide a timely, enlightening and systematic review on dissecting STIM functional domains, isoforms/variants, mechanisms of self-activation and gating of ORAI channels in mammals. In particular, the authors provided a in-depth discussions and crystal clear illustration for STIM1 C-terminus autoinhibitions and activations mechanisms by including recent structural information. 

I just have some minor points that can be very easily addressed by minor editing.

  1. Please add a brief introduction of (patho)physiological roles of STIM1 in the introductions part to educate the health relevance of the STIM1 related pathways.

  1. In part 3. regarding STIM N-Terminus: other than discussing the controversial binding ratio of STIM-N with Ca2+, the well-accepted Ca2+ binding sites and binding affinities (STIM1/STIM2) should also mentioned or commented.

  1. In part 5. Differences between STIM1 and STIM2.2. The author could further cite a recent research (PMID: 30444880), which systematically compares the molecular determinants between STIM1 and STIM2 in cellulo and in situ.

  1. P3, Line 100, one typo spotted: "canoncical" in "a canoncical EF hand (cEF)".

Author Response

Reviewer #1:

Herwig et al., provide a timely, enlightening and systematic review on dissecting STIM functional domains, isoforms/variants, mechanisms of self-activation and gating of ORAI channels in mammals. In particular, the authors provided a in-depth discussions and crystal clear illustration for STIM1 C-terminus autoinhibitions and activations mechanisms by including recent structural information.

I just have some minor points that can be very easily addressed by minor editing.

We thank the reviewer for the very positive reviewing comments.

  1. Please add a brief introduction of (patho)physiological roles of STIM1 in the introductions part to educate the health relevance of the STIM1 related pathways.

The introduction has been expanded accordingly (lines 55-64).

  1. In part 3. regarding STIM N-Terminus: other than discussing the controversial binding ratio of STIM-N with Ca2+, the well-accepted Ca2+ binding sites and binding affinities (STIM1/STIM2) should also mentioned or commented.

The distinct binding affinities of the STIM homologs (lines 124-126) as well as the established Ca2+ binding site within the EF-SAM domain (lines 131-134) have been included in the text.

  1. In part 5. Differences between STIM1 and STIM2.2. The author could further cite a recent research (PMID: 30444880), which systematically compares the molecular determinants between STIM1 and STIM2 in cellulo and in situ.

We now cite this publication (reference No. 92) and also added a brief description of the main findings to the text (lines 388-393).

  1. P3, Line 100, one typo spotted: "canoncical" in "a canoncical EF hand (cEF)".

The typo has been corrected (now line 109).

Reviewer 2 Report

This is a very comprehensive and up-to-date review onthe structure-function of STIM1.

Organisation and writing of the manuscripts are well done.

I would have appreciated more informations on interactions of the different STIM isoforms with other partners such as the intracellular calcium pumps...

I still wonder whether STIM, found in the plasma membrane, has any function given the high and constant concentration of extracellular calcium?

Author Response

Reviewer #2:

This is a very comprehensive and up-to-date review on the structure-function of STIM1.

Organisation and writing of the manuscripts are well done.

We thank the reviewer for the very positive reviewing comments.

I would have appreciated more informations on interactions of the different STIM isoforms with other partners such as the intracellular calcium pumps...

Thank you for this interesting suggestion. Another review by Berlansky et al. within the “STIMulating Ca2+ Homeostasis” special issue of IJMS already details the large family of STIM- and Orai-associated proteins, including intracellular Ca2+ pumps. We thus omitted them from our manuscript to avoid significant overlaps.

I still wonder whether STIM, found in the plasma membrane, has any function given the high and constant concentration of extracellular calcium?

The function of STIM in the plasma membrane is indeed an exciting topic and we reference several related studies (PMIDs: 10575208, 16537481, 23395841). However, most studies on PM-localized STIM were published many years ago and elaborate reviews detailing their contents are already available. With our present manuscript, we therefore put the focus on discussing recently published literature covering STIM functional domains and isoforms.